# In-Image Neural Machine Translation with Segmented Pixel Sequence-to-Sequence Model

**Yanzhi Tian[1]    Xiang Li[2]    Zeming Liu[3]    Yuhang Guo[1]\*    Bin Wang[2]**

[1]School of Computer Science and Technology, Beijing Institute of Technology, Beijing, China
[2]Xiaomi AI Lab, Beijing, China
[3]School of Computer Science and Engineering, Beihang University, Beijing, 100191, China
tianyanzhi@bit.edu.cn    lixiang21@xiaomi.com    zmliu@buaa.edu.cn
guoyuhang@bit.edu.cn    wangbin11@xiaomi.com

## Abstract

In-Image Machine Translation (IIMT) aims to convert images containing texts from one language to another. Traditional approaches for this task are cascade methods, which utilize optical character recognition (OCR) followed by neural machine translation (NMT) and text rendering. However, the cascade methods suffer from compounding errors of OCR and NMT, leading to a decrease in translation quality. In this paper, we propose an end-to-end model instead of the OCR, NMT and text rendering pipeline. Our neural architecture adopts an encoder-decoder paradigm with segmented pixel sequences as inputs and outputs. Through end-to-end training, our model yields improvements across various dimensions, (i) it achieves higher translation quality by avoiding error propagation, (ii) it demonstrates robustness for out domain data, and (iii) it displays insensitivity to incomplete words. To validate the effectiveness of our method and support for future research, we construct our dataset containing 4M pairs of De-En images and train our end-to-end model. The experimental results show that our approach outperforms both cascade method and current end-to-end model.[1]

## 1 Introduction

Machine Translation (MT) is an important technique for achieving communication between different languages (Zhang and Zong, 2020). Recently researches on MT mainly focus on text and speech modality (Fang et al., 2022; Fang and Feng, 2023), and there is less work based on image modality MT. In-Image Machine Translation (IIMT) is an **image-to-image** MT task which aims to transform images containing texts from one language to another. The technique of IIMT enables visual text translation across different languages, which helps users to

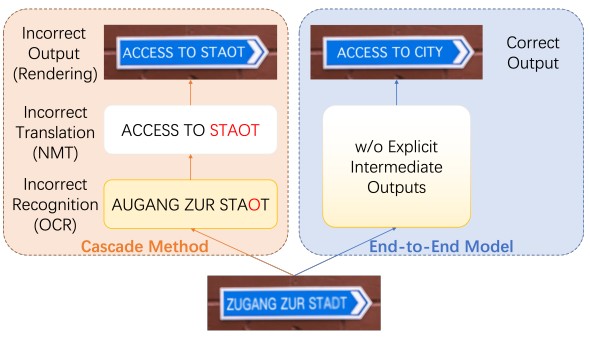

Figure 1: The illustration of two main architectures of IIMT. We observe that even minor error in the OCR output can negatively affect the subsequent translation quality for the cascade IIMT method.

understand street signs, scanned documents without the input of text manually (Mansimov et al., 2020). Compared with previous researches on image modality MT (Salesky et al., 2021; Ma et al., 2022; Lan et al., 2023), the most significant distinction lies in the outputs of IIMT which are images containing translated texts, making users understand the meaning of images more easily.

IIMT is a novel machine translation task, with inherent research value and a wide range of application scenarios. An intuitive method of IIMT is decomposing the task into several subtasks: firstly, recognizing the text in the image, then translating the text, and rendering the translated text into the image, which is commonly referred to as the cascade method. Specifically, the cascade method of IIMT usually contains three parts: (1) Recognizing: recognize the text in the origin image with an optical character recognition (OCR) model. (2) Translating: translate the text recognized by OCR model to the target language with a neural machine translation (NMT) model. (3) Rendering: eliminate text content in the original image and render the translated text.

However, a disadvantage of most cascade methods is the issue of error propagation. If errors occur

---

\*Corresponding Author.

[1]The code is available at https://github.com/YanzhiTian/E2E-IIMT

in the output of the previous stage model, it can directly impact the performance of the following models. As shown in Figure 1, "STA**D**T" in the image is incorrectly recognized as "STA**O**T" by the OCR model, and the NMT model incorrectly translates it. Consequently, the incorrect translated text is rendered into the image, resulting in the incorrect output image. Besides, some of the texts in the images may be incomplete because of the blocked by obstacles, resulting in the OCR model recognizing incomplete texts, thereby leading to the decreasing quality of NMT translations and the output images.

One of the methods to reduce the influence of error propagation and incomplete texts is designing end-to-end models instead of cascade methods. End-to-end models take into account the entire task directly and learn the transformation from input images to output images without explicit intermediate outputs, eliminating the need for intermediate steps or stages. Building end-to-end IIMT models is full of challenge because it is hard to generate images with specific texts (Ma et al., 2023), and few IIMT dataset is publicly available. Mansimov et al. (2020) propose end-to-end models for IIMT firstly. But the translation quality of their end-to-end models is much lower than that of cascade methods.

In this paper, we aim to develop an end-to-end IIMT model which achieves better translation quality than the cascade method. We regard images as pixel sequences, transforming the image-to-image generating task to a pixel sequence-to-sequence translation task and applying sequence-to-sequence architecture to build our end-to-end model. We also build IIMT datasets for training and validating our IIMT models.

The main contributions of this paper are as follows:

- To the best of our knowledge, we regard the image as a pixel sequence, transforming IIMT into a pixel sequence-to-sequence translation task first.

- We propose an end-to-end model based on segmented pixel sequences for IIMT with an improvement in translation quality compared with current end-to-end models and cascade methods.

- To facilitate the study of IIMT, we construct IIMT datasets to train and validate models, which can also serve as valuable resources for future research of IIMT.

## 2 Task Formulation

IIMT is a machine translation task where both input and output are images containing texts,

$$IIMT: \quad x \in \mathbb{R}^{H \times W \times C} \Rightarrow y \in \mathbb{R}^{H' \times W' \times C}, \tag{1}$$

where $x$ is the input image and $y$ is the output image, conforming

$$\hat{Y} = \arg\max_Y P(Y|X), \tag{2}$$

where $X$ is the text in the image $x$ and $Y$ is the translated text in the image $y$, and $\hat{Y}$ is the text in the decoded image.

## 3 Data Construction

To our knowledge, there is no publicly available dataset for IIMT. Since collecting real image pairs with aligned parallel texts for IIMT is costly, it is more practical to generate image pairs by rendering the texts from a parallel corpus. Therefore, we build datasets containing images with black font and white background of one-line text.

Specifically, we render texts in the parallel corpora into images with white background[2] to build the image pairs, replacing the text pair $\langle X, Y \rangle$ in parallel corpus to the image pair $\langle x, y \rangle$. For an image $x \in \mathbb{R}^{H \times W \times C}$, we set height $H = 32$, and width $W$ is proportional to the length of the text $X$. An image pair that contains parallel German-English (De-En) text in the IIMT dataset is shown in Figure 2.

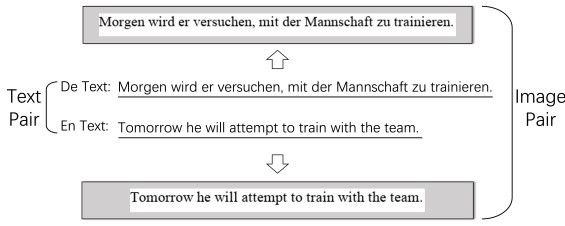

Figure 2: An example of image pair in our IIMT dataset.

Our method can be used to construct IIMT datasets for existing parallel corpora, making it easier to expand data. Besides, the font of texts in images can be changed by changing the font

---

[2]We render texts into images using the Python library `Pillow`, and the font for De and En texts is Times New Roman.

file, and the size or color of texts in images can be changed by adjusting parameters in the program, which means our method has strong scalability.

We perform inspections on the constructed datasets to ensure the quality, specifically verifying the integrity of text within the images and ensuring the absence of garbled characters.

# 4 Method

## 4.1 Model Architecture

Transformer (Vaswani et al., 2017) is a widely used sequence-to-sequence model in machine translation. Recent researches (Dosovitskiy et al., 2021; Liu et al., 2021) demonstrates the efficacy of the Transformer model in the field of computer vision. These findings demonstrate the capability of the Transformer model for the IIMT task. We use a vanilla Transformer as the backbone to build an end-to-end IIMT model as shown in Figure 3.

We regard images as segmented pixel sequences for IIMT. In our end-to-end IIMT model, the input image is converted into a pixel sequence, and then segmentation on the pixel sequence is applied to build a token sequence, which is converted to embeddings by the embedding layer. The embedding of the token sequence is the input of the Transformer encoder. Another token sequence is generated by auto-regressive decoding of the Transformer decoder, and then the sequence is recovered to the output image.

## 4.2 Image-To-PixelSequence

The **I**mage-**T**o-Pixel**S**equence (**ITS**) refers to a procedure of converting images into pixel sequences which is an image tokenization method:

$$F_{ITS}(x): \quad x \in \mathbb{R}^{H \times W \times C} \Rightarrow x_{tok} \in \mathbb{R}^{T \times 1}, \tag{3}$$

where $x$ is the input image with shape $H \times W \times C$, and $x_{tok}$ is the pixel sequence of $x$ with length $T$.

Given input image $x$ and output image $y$, after applying ITS, we can regard the IIMT task as a sequence-to-sequence translation task:

$$P(y_{tok}|x_{tok}, \theta) = \prod_{i=1}^{|y_{tok}|} p(y_{tok}^i|x_{tok}, y_{tok}^{<i}, \theta), \tag{4}$$

where $x_{tok} = F_{ITS}(x)$, $y_{tok} = F_{ITS}(y)$ and $\theta$ is the model parameters.

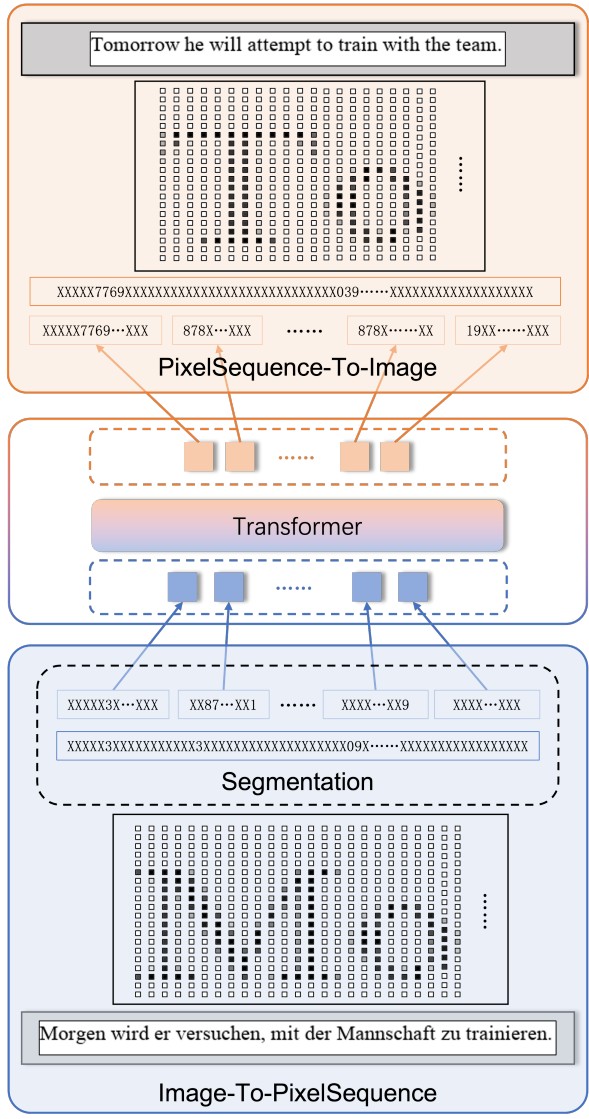

Figure 3: The architecture of our end-to-end IIMT model, including ITS, Transformer, and STI. The image is converted into a pixel sequence according to ITS, and segmented to the token sequence which is the input of the Transformer. The output of the Transformer is another token sequence that can be converted into the image with target text by STI.

The first step of ITS is converting an RGB image $x \in \mathbb{R}^{H \times W \times C}$ into a grayscale map[3] $x_g \in \mathbb{R}^{H \times W \times 1}$, and each value in the grayscale map is a float number ranging between 0 and 1. If each float number is regarded as a token, the type of token is too much. To solve this problem, we can quantize them into several discrete buckets and each bucket is represented by a character. A value $v$ in the grayscale map is converted into a character $c$

---

[3]Converted by transforms.Grayscale() of torchvision package.

with the following equation:

$$c = char(\lfloor \frac{v}{0.1} \rfloor) \quad (5)$$

The function $char()$ refers to converting the integer to a character. For example, the integer 1 is converted to "1", and the integer 2 is converted to "2" ...... and the integer 10 is converted to "X". After applying this method, the grayscale map is converted into a character map, and arrange the character map into a sequence of length $T$, where $T = H \times W$.

### 4.3 PixelSequence-To-Image

The Pixel**S**equence-**T**o-**I**mage (**STI**) refers to a procedure of converting pixel sequences into images which is an image detokenization method:

$$F_{STI}(x_{tok}): \quad x_{tok} \in \mathbb{R}^{T \times 1} \Rightarrow x \in \mathbb{R}^{H \times W \times C} \quad (6)$$

The output of the Transformer decoder is also a sequence decoded by auto-regressive which can be recovered to the image using STI.

Assume the length of the output sequence is $L$, and remove several characters from the end of the sequence so that its length $L'$ is divisible by $H$. Reshape the output sequence to character map with shape $H \times W$ where $W = \frac{L'}{H}$, and recover the character map to the grayscale map. A character $c$ in the character map is converted into a grayscale value $v$ with the following equation:

$$v = int(c) \times 0.1 \quad (7)$$

The function $int()$ refers to converting the character to an integer. For example, "1" is converted to 1, "2" is converted to 2 ...... and 'X' is converted to 10. Following this process, the character map is converted to a grayscale map. For each value in the grayscale map $v$, replace it with the RGB value of $(v \times 255, v \times 255, v \times 255)$. And finally get the output image $y \in \mathbb{R}^{H \times W \times C}$.

### 4.4 PixelSequence Segmentation

For an image $x \in \mathbb{R}^{H \times W \times C}$, the length of the pixel sequence is $H \times W$ which is much larger than the length of the token sequence of text-to-text MT. The cost of the self-attention in a Transformer for a sequence of length $N$ has $O(N^2)$ complexity, and the auto-regressive decoding also requires $N$ steps to generate the total output sequence. Which means the computational effectiveness is directly impacted by the sequence length.

We conduct segmentation on pixel sequences to build token sequences which can decrease the sequence length. One of the most widely used segmentation methods is BPE (byte pair encoding) (Sennrich et al., 2016). At each iteration of the BPE algorithm, the most frequent bi-gram in the sequences is merged together to build new token sequences. Generally, more iteration times correspond to a larger granularity and shorter token sequence.

We explore the use of BPE to pixel sequences and build token sequences which is called pixel sequence segmentation. Specifically, the most frequent appearance characters in the sequence will be merged at each iteration, which can increase the granularity and reduce the sequence length. After applying segmentation to pixel sequences with several times of iterations, the length of token sequences is close to text-to-text machine translation, which will have a similar computational complexity.

## 5 Experiments

### 5.1 Datasets

**De-En IIMT Dataset** We build an IIMT dataset (4M image pairs) with WMT14 De-En parallel corpus[4] to train our end-to-end IIMT model. The parallel corpus is filtered by the clean-corpus-n.perl of mosesdecoder[5] with a ratio 1.5, min 1 and max 250. Two in-domain test sets are built by newstest-2013 (3,000 image pairs) and newstest-2014 (3,003 image pairs).

In addition, we build two out-of-domain test sets to evaluate the domain adaption ability of our end-to-end model. We use tst-COMMON[6], which includes ASR transcriptions and translations of TED speech to build test sets in the spoken language domain (2,581 image pairs). We also use the Himl 2017 test set[7] which includes the health information texts and translations from NHS 24 and Cochrane to build test sets in the biomedical domain (1,512 image pairs).

### 5.2 Metrics

Since the outputs of IIMT models are images, it is hard to evaluate the translation quality with images directly. In order to employ evaluation metrics for

---

[4]https://www.statmt.org/wmt14
[5]https://github.com/moses-smt/mosesdecoder
[6]https://ict.fbk.eu/must-c-release-v2-0/
[7]https://www.himl.eu/test-sets

| Systems | In-Domain | | | | Out-Domain | | | |
|---|---|---|---|---|---|---|---|---|
| | newstest-2013 | | newstest-2014 | | tst-COMMON | | Himl | |
| | BLEU | COMET | BLEU | COMET | BLEU | COMET | BLEU | COMET |
| Cascade | 27.1 | 78.3 | 27.3 | 75.8 | 30.1 | 79.7 | 34.3 | 80.7 |
| Our E2E | **28.1** | **81.9** | **28.2** | **80.4** | **30.7** | **83.1** | **36.0** | **84.3** |

Table 1: Experimental results on in-domain (news) and out-domain (spoken language and biomedical) IIMT datasets.

| Systems | BLEU |
|---|---|
| Conv Baseline (Mansimov et al., 2020) | 0.5 |
| AttnConv (Mansimov et al., 2020) | 7.7 |
| Our E2E | **28.1** |

Table 2: Experimental results of end-to-end models on newstest-2013 IIMT dataset.

machine translation, we use OCR model to recognize text in the image, and evaluate the translation quality with the output of OCR model and reference.

### 5.3 Experimental Settings

We utilize the OCR model in PaddleOCR[8] which is a commonly-used OCR toolkit containing models of different languages to recognize texts in the generated images. We compute BLEU score (Papineni et al., 2002) by SacreBLEU[9] (Post, 2018) and COMET[10] (Rei et al., 2020) to evaluate translation quality.

We conduct experiments on the following systems:

**Conv Baseline** An end-to-end model for IIMT (Mansimov et al., 2020) which is based on UNet architecture.

**AttnConv** An end-to-end model for IIMT (Mansimov et al., 2020) which contains a Covolutional Encoder, a Convolutional Decoder, and a Self-Attention Encoder.

**Cascade** The IIMT cascade method, including an OCR model, an NMT model, and text rendering, is the same as the cascade method in Figure 1. We use the OCR model in the PaddleOCR toolkit to recognize German texts in images. The NMT model is Transformer Big, trained with the same parallel corpora which are used to construct the IIMT dataset. Both German and English texts are applied 32K BPE. Beam search is applied to the decoding stage of the NMT model, and the beam

[8] https://github.com/PaddlePaddle/PaddleOCR
[9] https://github.com/mjpost/sacrebleu
[10] https://github.com/Unbabel/COMET

size is 5.

**Our E2E** Our end-to-end IIMT model which is introduced in Section 4. We apply the Transformer Big model implemented by Fairseq toolkit (Ott et al., 2019) to build our end-to-end model. We use the training set to learn the order of pixel sequence segmentation and apply the segmentation to the total dataset. The iteration times for both pixel sequences converted from German and English images is 50,000. The token sequence is decoded by beam search, and the beam size is 5.

### 5.4 Main Results

The experimental results are shown in Table 1, including the BLEU score and COMET of in-domain and out-domain datasets. On both in-domain and out-domain datasets, the performance of our end-to-end model is better than the cascade method. This phenomenon is mainly caused by error propagation of the cascade method, especially the error of the OCR model.

Besides, the translation quality of our end-to-end model has an improvement compared with the current end-to-end models (Conv baseline, AttnConv) (Mansimov et al., 2020), which is shown in Table 2. The reason why our end-to-end model performs better is mainly that our model uses ITS and STI, converting images into pixel sequences and applying pixel sequence segmentation to enhance the utilization of textual information within images. But the current end-to-end models can not fully use the textual information in the image.

### 6 Analysis

In this section, we focus on the following research questions (RQs): (1) Does the segmentation of different iterations influence the translation quality? (2) Compared to the cascade method, does our model has better translation quality for images with incomplete texts? (3) Does the position of texts in images influence the performance of our model? (4) Is our method effective on datasets with multiple font sizes or font types? (5) How does error

propagation influence the translation quality of the cascade method?

## 6.1 RQ1: Translation Quality of Different Segmentation

We conduct different segmentation to pixel sequences, building token sequences with different lengths and granularity. Figure 4 shows the relationship between the iteration times of pixel sequence segmentation and the BLEU score. The results demonstrate that with the increasing iteration times of segmentation, the length of the token sequence is decreased, and the granularity of the token is increased.

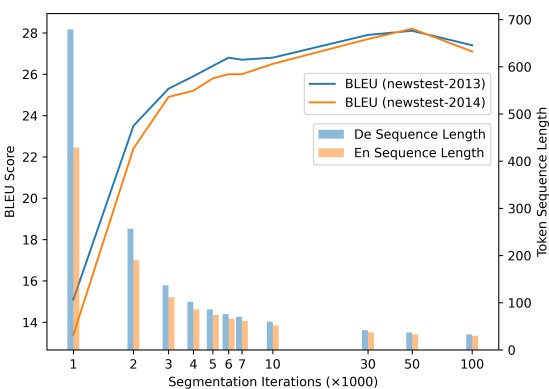

Figure 4: The relationship between BLEU score and iteration times of segmentation. Larger segmentation iterations refers to a larger granularity.

We also find that pixel sequences representing images containing specific characters or subwords will be merged together as segments after several iterations, and different iteration times corresponding to different segmentation, as shown in Figure 5. Therefore, the token contains more textual information with clearer semantic boundaries after more iteration times of segmentation, leading to better translation quality.

Besides, with the decreasing length of the token sequence, the decoding time is firstly decreased. However, too many iteration times cause a larger token vocabulary, with more parameters in the model, which may cause a longer decoding time. The numerical experimental results are shown in Appendix A.

## 6.2 RQ2: Incomplete Texts in Images

Some of the texts in the images from the real world may be incomplete because of the occlusion of

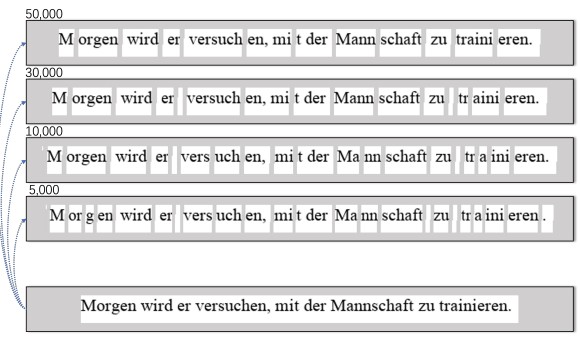

Figure 5: Visualization of segmentation with different iteration times. The iteration times for each segmentation are indicated in the upper-left corner. With the increasing of iteration times, the granularity of the token is increased and the length of the token sequence is decreased.

obstacles. We simulate this type of data to build test sets with images of incomplete texts.

We randomly mask $20 \times 20$ for each image in the origin test sets to build new test sets with incomplete images, and Figure 6 shows an example of incomplete image. Experimental result in Table 3 shows that our end-to-end model has a better translation quality compared with the cascade method on this type of test set. The result is benefiting to our approach of converting images as pixel sequences. Since there is no need to explicitly recognize texts within images, our model is less affected by the problem of incomplete texts.

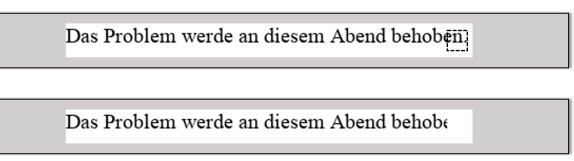

Figure 6: An example of an incomplete image. The figure above represents the original image, while the figure below is the image after masking the areas within the dashed box of the original image.

Incomplete texts in images affect the OCR outputs in the cascade method and directly impact the subsequent results. We conduct experiments to calculate the WER (word error rate)[11] of the OCR model for each type of images. For the origin images, the WER of newstest-2013 and newstest-2014 are 1.6% and 1.4%. For the incomplete images, the WER of newstest-2013 and newstest-2014 are 3.7% and 3.6%.

---

[11]Computed by `nltk.edit_distance()`

| Types | Systems | newstest-2013 | | newstest-2014 | | tst-COMON | | Himl | |
|---|---|---|---|---|---|---|---|---|---|
| | | BLEU | COMET | BLEU | COMET | BLEU | COMET | BLEU | COMET |
| Origin | Cascade | 27.1 | 78.3 | 27.3 | 75.8 | 30.1 | 79.7 | 34.3 | 80.7 |
| | Our E2E | **28.1** | **81.9** | **28.2** | **80.4** | **30.7** | **83.1** | **36.0** | **84.3** |
| RQ2 | Cascade | 24.2 | 71.2 | 24.5 | 69.1 | 26.4 | 71.7 | 30.5 | 72.4 |
| | Our E2E | **25.0** | **72.3** | **24.7** | **71.6** | **26.9** | **72.1** | **31.1** | **73.6** |
| RQ3 | Cascade | 27.1 | 78.3 | 27.3 | 75.8 | 30.1 | 79.7 | 34.3 | 80.7 |
| | Our E2E | **27.8** | **81.5** | **27.8** | **80.0** | **30.5** | **82.7** | **35.5** | **84.1** |

Table 3: Experimental results on different types of test sets, including the origin test sets, the test sets with incomplete texts in images (RQ2) and the test sets with different positions of texts in images (RQ3).

### 6.3 RQ3: Influence of Texts Position in Images

Since the OCR model usually contains a text detection module, the cascade method is not sensitive to the position of texts in images. However, our end-to-end model does not conduct OCR and text detection explicitly.

To validate whether our model is influenced by the texts position in images, we render texts with a shifting position compared with the origin test sets, and Figure 7 shows an example of an image containing text with shifting position.

Das Problem werde an diesem Abend behoben.

Das Problem werde an diesem Abend behoben.

Figure 7: An example of an image with shifting position of text. The figure above represents the original image, while the text in the figure below is vertically and horizontally shifting by 5 pixels.

As the experimental results shown in Table 3, although there is a certain decrease in translation quality, it is still obtaining better translation quality compared with the cascade method. This result is attributed to our ITS method. For the pixel sequence of an image with a different position of text, it is equivalent to adding a prefix with character "X" representing several white pixels, and the length of the prefix is dependent on the shifting distance of text. This means the rest of the pixel sequence remained unchanged and different position of text in the image influences ITS slightly, leading to the minimal decrease of translation quality.

### 6.4 RQ4: Mutiple Font Sizes and Font Types

We also construct IIMT datasets with multiple font sizes and font types, as shown in Figure 8. The

TNR15 is constructed with Times New Roman, and the font size is 15. The TNR25 is constructed with Times New Roman, and the font size is 25. The Arial20 is constructed with Arial, and the font size is 20.

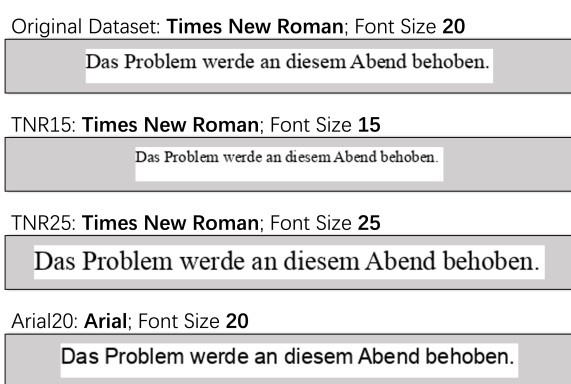

Figure 8: Datasets with multiple font sizes and font types.

We train several models with these datasets, and compute the BLEU score with the same method in Section 5.2. The BLEU scores are shown in Table 4. The BLEU scores of TNR25 and Arial20 are similar to the results in Table 1, indicating that our method is still effective on datasets with multiple font sizes and font types.

| Datasets | newstest-2013 | newstest-2014 |
|---|---|---|
| TNR15 | 22.3 | 22.6 |
| TNR25 | 27.5 | 27.7 |
| Arial20 | 27.9 | 28.1 |

Table 4: BLEU scores of models trained on different datasets.

However, the BLEU score of TNR15 is much lower than that in Table 1, and the reason is that the WER of the target-side OCR model is higher while recognizing English texts with smaller font

| Systems | newstest-2013 | | newstest-2014 | |
|---|---|---|---|---|
| | BLEU | COMET | BLEU | COMET |
| Golden NMT | 29.8 | 83.0 | 29.7 | 81.5 |
| Cascade | 27.1 (-2.7) | 78.3 (-4.7) | 27.3 (-2.4) | 75.8 (-5.7) |
| | tst-COMMON | | Himl | |
| | BLEU | COMET | BLEU | COMET |
| Golden NMT | 33.0 | 84.0 | 37.8 | 84.7 |
| Cascade | 30.1 (-2.9) | 79.7 (-4.3) | 34.3 (-3.5) | 80.7 (-4.0) |

Table 5: Comparison of translation quality of golden NMT (w/o OCR error) and cascade method.

size. The WER of newstest-2013 and newstest-2014 with font size 20 (same setting as Table 1) are 1.4% and 1.6%. The WER of newstest-2013 and newstest-2014 with font size 15 are 4.2% and 4.0%.

## 6.5 RQ5: Error Propagation of Cascade Method

In our experiments, we observe that the OCR model may recognize some characters incorrectly. For example, the character 'ü' may be recognized as 'u' and 'ß' as 'B' because of their similar shapes. The incorrectly recognized results are transferred to following stages, leading to error propagation of the cascade method.

To analyze the influence of error propagation, we first use the OCR model to recognize images containing German texts in the datasets, and then calculate WER of recognized results. For images built by newstest-2013 and newstest-2014, the WER is 1.6% and 1.4% separately.

We also compute the BLEU score and COMET of the NMT model using the origin text as input (namely golden NMT), which is without errors caused by the OCR model. In order to mitigate the influence of OCR errors on the evaluation of the cascade method, we render the output of the golden NMT into images with the same method and recognize the texts using the same OCR. As shown in Table 5, both the BLEU score and COMET of the cascade method are lower than the golden NMT because of the errors in the outputs of the OCR model.

The reason why errors in OCR outputs influence the performance of the NMT model obviously is because the OCR model recognizes the text with character level which means there may be an error occurring with one letter in a word while other letters in the same word are correct. It will result in the word with the wrong letter and the tokenization

method on the text such as BPE will give wrong subwords, which will damage the performance of NMT a lot.

An example of error propagation in the cascade method is shown in Table 6. The OCR model recognizes the word "außerdem" as "auBerdem", and the BPE result changes from "außerdem" to "au@@ Ber@@ dem", resulting in the wrong translation "Berdem".

The result of our end-to-end model with the same input image is shown in Table 7, and we can observe that our end-to-end model mitigates the problem of error propagation in the cascade method.

## 7 Related Work

### 7.1 In-Image Machine Translation

There are mainly two difficulties to build an end-to-end model for IIMT: Firstly, there is no dataset for IIMT publicly available. Secondly, although there are several researches on image synthesis such as UNet (Ronneberger et al., 2015), GAN (Goodfellow et al., 2014), VAE (Kingma and Welling, 2022), Diffusion Models (Ho et al., 2020), and the Stable Diffusion Model (Rombach et al., 2022) has made great progress in generating high-resolution images, all of them have limitation on generating text within images. Liu et al. (2022) and Ma et al. (2023) focus on image synthesis with characters, and achieve an improvement compared with the previous models. But there is still a certain gap to implement end-to-end IIMT.

Current end-to-end models (Mansimov et al., 2020) include a Conv baseline and an AttnConv. The Conv baseline is based on UNet architecture, which can only generate the image with one or two words, and it quickly devolves into noise. The AttnConv contains a Convolutional Encoder, a Convolutional Decoder, and a Self-Attention Encoder.

| | Cascade Method (w/ OCR error) | Golden (w/o OCR error) |
|---|---|---|
| Input Image | Die Täter schossen außerdem auf eine Bushaltestelle. | |
| OCR | Die Täter schossen auBerdem auf eine Bushaltestelle. | Die Täter schossen außerdem auf eine Bushaltestelle. |
| BPE | Die Täter scho@@ ssen au@@ Ber@@ dem auf eine Bushaltestelle . | Die Täter scho@@ ssen außerdem auf eine Bushaltestelle . |
| NMT | The perpetrators shot Berdem on a bus stop . | The perpetrators also shot at a bus stop . |
| Output Image | The perpetrators shot Berdem on a bus stop . | |

Table 6: Example of error propagation of cascade method, the golden result is generated by NMT model using the original text (w/o OCR error).

| Input Image | Die Täter schossen außerdem auf eine Bushaltestelle. |
|---|---|
| Output Image of Cascade Method | The perpetrators shot Berdem on a bus stop . |
| Output Image of Our End-to-End Model | The perpetrators also shot a bus stop. |

Table 7: Comparison of the output of the cascade method and our end-to-end model.

## 7.2 Text Image Translation

Text Image Translation (TIT) aims to translate images containing source language texts into the target language, which is an image-to-text machine translation task. Compared with IIMT, TIT is a multimodal machine translation (MMT) task (Elliott et al., 2016; Calixto et al., 2017; Ive et al., 2019; Zhang et al., 2020). TIT primarily contains two categories of methods, namely cascade methods including OCR and NMT, and end-to-end models.

Cascade methods of TIT also suffer from error propagation, one of the methods is to associate images with relevant texts, providing useful supplementary information for translation (Lan et al., 2023). Another method is to design end-to-end models (Ma et al., 2022) which mainly contain vision encoder such as ResNet (He et al., 2016) and text decoder.

## 7.3 Byte Pair Encoding (BPE)

BPE (Sennrich et al., 2016) mainly contains two stages, learning BPE and applying BPE. During learning BPE, the tokens will be split into the smallest unit, and the most frequent bi-gram is merged as a new token. This process usually involves multiple iterations and finally results in a BPE code list that contains $n$ pairs of bi-gram after $n$ iterations. During applying BPE, the tokens will also

be split into the smallest unit and the bi-gram will be merged according to the BPE code list in turn. Recent work explores the use of the BPE technique to image modality (Razzhigaev et al., 2022). They apply BPE on pixel sequences of iGPT (Chen et al., 2020), increasing the performance and efficiency.

## 8 Conclusion

In this paper, we propose an end-to-end model for IIMT based on segmented pixel sequences to reduce the influence of error propagation. Experimental results on IIMT datasets show that our model has better translation quality compared with both the current end-to-end model and the cascade method. Furthermore, we conduct analysis to demonstrate our end-to-end model has better performance on different types of IIMT datasets. In future work, we will try to build more realistic IIMT datasets, and design new model architecture to achieve a better translation quality.

## Limitations

While in-image machine translation is a new MT task, and our end-to-end model outperforms the previous methods, this work has certain limitations. Firstly, our work is a preliminary study on in-image machine translation (e.g. images with black font and white background, converting RGB images to

grayscale maps). And the Image-To-PixelSequence phase of our end-to-end model requires a learning segmentation method based on pixel sequences, thereby demanding substantial CPU computational resources. Besides, the application of the segmentation method also requires a certain computational cost. Consequently, there is no improvement in decoding time compared to the cascade methods.

Furthermore, it is hard to combine our method with vision models. The input images of recent vision models (e.g. Vision Transformer) are fixed size, but the images in our dataset are different sizes, which need to crop the images. However, the Vision Transformer is still worth trying. In the future, we will try to combine our method with vision models, and take both the original images and the OCR results as inputs.

## Ethics Statement

This paper proposes an end-to-end IIMT model. We take ethical considerations seriously and ensure that the methods used in this study are conducted in a responsible and ethical manner. Our IIMT datasets are built by publicly available parallel corpora, which are used to support research of IIMT.

## Acknowledgements

We thank all the anonymous reviewers for their insightful and valuable comments. This work is supported by the National Key R&D Program of China (No. 2020AAA0106600).

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

| Iterations | Avg. Len (De/En) | BLEU | | Decoding Time (s) | |
|---|---|---|---|---|---|
| | | newstest-2013 | newstest-2014 | newstest-2013 | newstest-2014 |
| 1,000 | 679/429 | 15.1 | 13.4 | 2935 | 3344 |
| 2,000 | 257/190 | 23.5 | 22.4 | 1094 | 1074 |
| 3,000 | 137/112 | 25.3 | 24.9 | 566 | 647 |
| 4,000 | 102/86 | 25.9 | 25.2 | 449 | 476 |
| 5,000 | 86/74 | 26.4 | 25.8 | 368 | 456 |
| 6,000 | 76/66 | 26.8 | 26.0 | 334 | 323 |
| 7,000 | 70/61 | 26.7 | 26.0 | 311 | 320 |
| 10,000 | 60/52 | 26.8 | 26.5 | 260 | 266 |
| 30,000 | 42/37 | 27.9 | 27.7 | 244 | 246 |
| 50,000 | 37/33 | 28.1 | 28.2 | 187 | 201 |
| 100,000 | 33/30 | 27.4 | 27.1 | 214 | 211 |

Table 8: Numerical experimental results of RQ1.

# A   Numerical Experimental Results of RQ1

The numerical experimental results are shown in Table 8, and the experiments of decoding time are conducted on 1 Tesla V100 GPU with batch 10.