# OpenReview forum: "In-Image Neural Machine Translation with Segmented Pixel Sequence-to-Sequence Model"
_EMNLP/2023/Conference — EMNLP 2023 Findings_

### Official Review · Reviewer_9Cpu · 2023-07-25

**Soundness:** 4

**Excitement:**

4: Strong: This paper deepens the understanding of some phenomenon or lowers the barriers to an existing research direction.

**Missing References:**

Robust Open­Vocabulary Translation from Visual Text Representations

**Paper Topic And Main Contributions:**

This paper focuses on a new task, In-Image Machine Translation (IIMT), and proposes a novel and effective solution. The input and output images are converted into pixel sequences, and the grayscale values are mapped to discrete symbols (1-10). A BPE vocabulary is learned on the symbol sequence, and a sequence-to-sequence Transformer model is used to learn the mapping between the source and target sequences. Experimental results on a created benchmark demonstrate the effectiveness of the proposed method.

**Questions For The Authors:**

- It is not entirely clear how an image with a size of HxW is converted into a sequence of length T in the proposed method. However, it appears that the image is organized as a sequence from top to bottom and then from left to right, as shown in Figure 5. More explanation is needed in the paper to clarify this process.
- The BLEU score of the Gloden NMT model on the newstest-2014 seems slightly higher than results reported in previous NMT paper. What about the details of your NMT model?

**Reasons To Accept:**

- The proposed framework is surprisingly effective in translating the source image to the target image without any textual supervision during training, outperforming the cascaded OCR+MT baseline. It demonstrates that the segmented pixel sequence is a simple yet effective intermediate representation for this task.
- The paper is well written and easy to follow. The authors did detailed analysis of the proposed method.

**Reasons To Reject:**

The created dataset primarily contains images with black text and white backgrounds, which may be simplified compared to real-world scenarios. It is unclear if the proposed method will outperform the cascaded method in more complex scenarios.

**Reproducibility:**

4: Could mostly reproduce the results, but there may be some variation because of sample variance or minor variations in their interpretation of the protocol or method.

**Reviewer Confidence:**

4: Quite sure. I tried to check the important points carefully. It's unlikely, though conceivable, that I missed something that should affect my ratings.

---

> ### Author Rebuttal · Authors · 2023-08-29
>
> Thank you for your detailed comments, and here are our responses.
>
> **Reason To Reject 1:**
>
> > The created dataset primarily contains images with black text and white backgrounds, which may be simplified compared to real-world scenarios. It is unclear if the proposed method will outperform the cascaded method in more complex scenarios.
>
> Our dataset is close to real-world scenarios that contain a lot of text, such as webpage screenshots and scanned documents. We have constructed datasets with different fonts and font sizes, and our end-to-end models reach similar results. We will add the results in our new version.
>
> To the best of our knowledge, there is no public available dataset for in-image machine translation, and our dataset serves as a good test-bed for initial attempts. Our work is a preliminary study on in-image machine translation, and we achieve better translation quality than cascade method in this scenario. We hope our method and dataset can facilitate the study of in-image machine translation, and there will be more further research.
>
> **Question 1:**
>
> > It is not entirely clear how an image with a size of HxW is converted into a sequence of length T in the proposed method. However, it appears that the image is organized as a sequence from top to bottom and then from left to right, as shown in Figure 5. More explanation is needed in the paper to clarify this process.
>
> We describe the conversion of an image into a sequence in sub-section 4.2, which may not be very detailed. The actual process is what you said. The grayscale map is firstly converted into the character map (Line 191-Line 200). Then the character map is organized as a sequence from top to bottom and then from left to right, and the character map with a size of HxW is converted into a sequence of length T (T=HxW) (Line 202-Line 203.) Thanks for your suggestion, we have described it more clearly in the new version to eliminate misunderstandings.
>
> **Question 2:**
>
> > The BLEU score of the Gloden NMT model on the newstest-2014 seems slightly higher than results reported in previous NMT paper. What about the details of your NMT model?
>
> We conduct experiments on newstest-2014 De-En (Line 262-Line 269), and the settings of Golden NMT is same as the NMT in the cascade method (Transformer Big, 32K BPE and beam 5, described at Line 318-Line 324).
>
> There are two translation directions of newstest-2014, including De-En and En-De. For De-En, the BLEU score with similar settings is 31.4 [1]. For En-De, the BLEU score is 28.1 [1].
>
> The BLEU score of De-En is higher than our Golden NMT (29.7). The reason is to make our systems comparable, we render the translated text obtained by Golden NMT into the image, and use the same OCR model to recognize the text (Line 442 - Line 446). Because of errors in OCR results, the BLEU score is lower than other work on newstest-2014 De-En.
>
> **Missing References:**
>
> > Robust Open­Vocabulary Translation from Visual Text Representations
>
> Thanks for your reminder, we have added it in our new version.
>
> Thanks for your comments to help us improve our work, and we will continue to polish the paper to make it better.
>
> [1] Rothe S, Narayan S, Severyn A. Leveraging pre-trained checkpoints for sequence generation tasks[J]. Transactions of the Association for Computational Linguistics, 2020, 8: 264-280.

---

### Official Review · Reviewer_6iiA · 2023-08-02

**Soundness:** 4

**Excitement:**

3: Ambivalent: It has merits (e.g., it reports state-of-the-art results, the idea is nice), but there are key weaknesses (e.g., it describes incremental work), and it can significantly benefit from another round of revision. However, I won't object to accepting it if my co-reviewers champion it.

**Paper Topic And Main Contributions:**

The paper focuses on in-image neural machine translation. Unlike previous cascade approaches that first perform OCR, then do text translation and text rendering, this paper constructs an image-to-image translation dataset and directly train an end-to-end translation model. They demonstrate that this can alleviate the error propagation issue and improve the overall translation performance.

**Reasons To Accept:**

1. Building end-to-end in-image machine translation models is technically sound and can alleviate the error propagation issue.
2. The constructed dataset can be helpful for future research.
3. The idea is simple and effective.

**Reasons To Reject:**

1. My main concern is that the constructed dataset is rather artificial and can have limited application scenarios. Specifically, they directly render texts into images with white background, which cannot well represent the diverse real-world scenarios.
2. Some experimental results can be questionable and do not provide readers with much insight. For example, I am not sure why would their method can be better at translating incomplete texts in images compared to pipeline methods.
3. It would be better to include a model that take both the original image and the OCR results as inputs.

**Reproducibility:**

3: Could reproduce the results with some difficulty. The settings of parameters are underspecified or subjectively determined; the training/evaluation data are not widely available.

**Reviewer Confidence:**

3: Pretty sure, but there's a chance I missed something. Although I have a good feel for this area in general, I did not carefully check the paper's details, e.g., the math, experimental design, or novelty.

---

> ### Author Rebuttal · Authors · 2023-08-29
>
> Thank you for your detailed comments, and here are our responses.
>
> **Reason To Reject 1:**
>
> > My main concern is that the constructed dataset is rather artificial and can have limited application scenarios. Specifically, they directly render texts into images with white background, which cannot well represent the diverse real-world scenarios.
>
> Our dataset is close to real-world scenarios that contain a lot of text, such as webpage screenshots and scanned documents. We have constructed datasets with different fonts and font sizes, and our end-to-end models reach similar results. We will add the results in our new version.
>
> To the best of our knowledge, there is no public available dataset for in-image machine translation, and our dataset serves as a good test-bed for initial attempts. Our work is a preliminary study on in-image machine translation, and we achieve better translation quality than cascade method in this scenario. We hope our method and dataset can facilitate the study of in-image machine translation, and there will be more further research.
>
> **Reason To Reject 2:**
>
> > Some experimental results can be questionable and do not provide readers with much insight. For example, I am not sure why would their method can be better at translating incomplete texts in images compared to pipeline methods.
>
> For images with incomplete texts, some words in the images are incomplete which will influence the output of OCR model, and we conduct experiments to calculate the WER of OCR model for each type of images. For the origin images, the WER for newstest-2013 and newstest-2014 are 1.6% and 1.4%. For the incomplete images, the WER for newstest-2013 and newstest-2014 are 3.7% and 3.6%. More errors in the OCR output influence the BPE output more seriously, which will damage the NMT performance a lot.
>
> For our end-to-end model, there is a decrease in translation quality, but it is less affected than the cascade method. The experiment results are shown in Table 3, RQ2. Thanks for your question, and we have added more detailed analysis at sub-section 6.4.
>
> **Reason To Reject 3:**
>
> > It would be better to include a model that take both the original image and the OCR results as inputs.
>
> Thanks for your suggestion, we will try to conduct experiments on this model, and compare with our method.
>
> Thanks for your comments to help us improve our work, and we will continue to polish the paper to make it better.

---

### Official Review · Reviewer_ApGL · 2023-08-02

**Soundness:** 2

**Excitement:**

2: Mediocre: This paper makes marginal contributions (vs non-contemporaneous work), so I would rather not see it in the conference.

**Missing References:**

Path-based image encoding should be considered as a vital baseline for this paper:
[1] Alexey Dosovitskiy, Lucas Beyer, Alexander Kolesnikov, Dirk Weissenborn, Xiaohua Zhai, Thomas Unterthiner, Mostafa Dehghani, Matthias Minderer, Georg Heigold, Sylvain Gelly, Jakob Uszkoreit, Neil Houlsby: An Image is Worth 16x16 Words: Transformers for Image Recognition at Scale. ICLR 2021


**Paper Topic And Main Contributions:**

This paper proposes an end-to-end image-to-image machine translation model. The main contribution of this paper is using a pixel sequence-based method to transform source language text images into target language text images.

**Questions For The Authors:**

1. In the task formulation section, what's the meaning of \hat{Y}? Although the readers might guess the meaning of \hat{Y}, the authors should clarify to eliminate misunderstandings.
2. The data construction in this paper just renders texts in the parallel corpora into images with white background, which is quite easy data distribution of text images. The authors mention their method has strong scalability, I wonder whether authors conduct any experiments on other data distribution of text images like changing the font, font colors, background colors, and background images?
3. What's the method of converting an RGB image into a grayscale map in your implementation (Line 186-188 in the paper)? There are multiple methods of converting an RGB image into a grayscale map, what's your implementation? PIL.image.convert(“L”) or pixel averaging through channels or any other methods? If the channel conversion is conducted using open-source packages or through existing methods, please provide the conversion formula, reference citation, or footnote with a reference link to the open-source package.
4. Since the dataset in your work is images with black font and white background of the one-line text. Is the RGB image necessary and do you conduct any ablation study?
5. Figure 5 shows the Visualization of segmentation with different iteration times. Each segmentation results have several missing pixels at the bottom-right corner, what’s the reason for these missing pixels? Furthermore, What about the robustness of the proposed segmentation method? When there are Gaussian noise, Salt-and-pepper noise, and other pixel noises, what are the influences of the segmentation results? Have you conducted any analysis experiments? Experiments and analysis should be conducted in the paper to show the generalization of the proposed pixel sequence segmentation.
6. In sub-section 6.4, there is no comparison and analysis of end-to-end models with cascade models. As so, error propagation indeed exists in cascade methods, and can your proposed end-to-end methods address this problem? What's the relationship between the analysis of error propagation of the cascade method with your proposed end-to-end method? Furthermore, the case study in Table 5 lacks the results of your proposed end-to-end model, which cannot prove your end-to-end model can reduce the influence of error propagation.
7. For image encoding and decoding, the pixel sequence transformation and pixel sequence segmentation might be one of your contributions. However, there is no comparison with vanilla patch-based image encoding and decoding methods, which is a quite vital baseline for your work.


**Reasons To Accept:**

The experiment results show the proposed method can outperform cascade model on in-domain and out-domain test sets.

**Reasons To Reject:**

This paper has several shortcomings that should be rejected:
1. The description of the pixel sequence-based in-image machine translation model is unclear, such as 1) how to transform the image into pixel sequence is incomplete, 2) the method of transforming RGB to grayscale map is unclear, and 3) the training and inference details are missing, and the optimization object is not mentioned.
2. The constructed dataset in this paper is quite simple, which just considers the black font and white background image. As so, the experiments and analysis are based on a toy task in the in-image translation area, which has limited influence in real-world applications. Various text image effects should be considered in ablation studies to show the generalization of the proposed methods.
3. There are certain problems with paper writing. Some explanations for variables and symbols are missing. Thus, this paper should take a major revision before publication.


**Reproducibility:**

4: Could mostly reproduce the results, but there may be some variation because of sample variance or minor variations in their interpretation of the protocol or method.

**Reviewer Confidence:**

4: Quite sure. I tried to check the important points carefully. It's unlikely, though conceivable, that I missed something that should affect my ratings.

**Typos Grammar Style And Presentation Improvements:**

What's the meaning of T in Equation (3)? Meanwhile, the detailed clarification of function F_{ITS}(x) is missing. Subsection 4.2 only introduces 1) transforming the RGB image into a grayscale map and 2) quantizing values in the grayscale map into a character map. I believe that there must have a step 3) transforming character map into pixel sequences but it is missing in the paper. If there is no step 3), the character map of length H\times W (Line 203 in the paper) is different from x_{tok}\in\mathbb{R}^{T\times 1} as in Equation (3).

---

> ### Author Rebuttal · Authors · 2023-08-29
>
> Thank you for your detailed comments, and here are our responses.
>
> **Reason To Reject 1:**
>
> > The description of the pixel sequence-based in-image machine translation model is unclear, such as 1) how to transform the image into pixel sequence is incomplete, 2) the method of transforming RGB to grayscale map is unclear, and 3) the training and inference details are missing, and the optimization object is not mentioned.
>
> 1)At Line 202 - Line 203, we describe the character map that is converted into a sequence of length HxW. We may not describe this process clearly. Actually, the character map is organized as a sequence from top to bottom and then from left to right, and the character map with a size of HxW is converted into the 1-D sequence of length 1xT (T=HxW).  Thanks for your suggestion, and we have described this step more clearly in the new version.
>
> 2)We do not introduce the detailed method. We use transforms.Grayscale() of torchvision package to convert the RGB image into the grayscale map. Thanks for your suggestion, and we have added it in the new version.
>
> 3)We may not describe it clearly at Line 304 - Line 313. Our method of transforming the image into pixel sequence modeling the image as sequence, and the widely used method of seq2seq task can be used, such as the model (Transformer Big), the loss function (Cross Entropy) and decoding method (Beam Search). Thanks for your suggestion, and we have added more detail at Line 304 - Line 313 in our new version.
>
> **Reason To Reject 2:**
>
> > The constructed dataset in this paper is quite simple, which just considers the black font and white background image. As so, the experiments and analysis are based on a toy task in the in-image translation area, which has limited influence in real-world applications. Various text image effects should be considered in ablation studies to show the generalization of the proposed methods.
>
> In-image machine translation is a new MT task, which requires images as outputs, exploring the modality of MT. And the research of end-to-end in-image machine translation is important, but there is few related work [1].
>
> To the best of our knowledge, there is no public available dataset for in-image machine translation, and our dataset serves as a good test-bed for initial attempts. Our work is a preliminary study on in-image machine translation, and we achieve better translation quality than cascade method in this task.
>
> Our dataset is close to real-world scenarios that contain a lot of text, such as webpage screenshots and scanned documents. We have constructed datasets with different fonts and font sizes, and our end-to-end models reach similar results. We will add the results in our new version.
>
> We hope our method and dataset can facilitate the study of in-image machine translation, and there will be more further research.
>
> **Reason To Reject 3:**
>
> > There are certain problems with paper writing. Some explanations for variables and symbols are missing. Thus, this paper should take a major revision before publication.
>
> Thanks for your suggestion, we have proofread and revised the manuscript thoroughly according to your constructive comments, including but not limited to:
>
> - We describe the variable more clearly in task formulation (Line 122).
> - We add the description of the variable x_{tok} of Equation 3 (Line 179).
> - We introduce the method of converting the RGB image into grayscale map (Line 187 - Line 188).
> - We fix the typo in Equation 5 (Line 196), which should be c=char(\lfloor \frac{v}{0.1} \rfloor).
> - We introduce the step of converting the character map to sequence in detail (Line 202 - Line 203).
> - We add the detailed explaination of Figure 5.
>
> **Question 1:**
>
> > In the task formulation section, what's the meaning of \hat{Y}? Although the readers might guess the meaning of \hat{Y}, the authors should clarify to eliminate misunderstandings.
>
> The definition of variable \hat{Y} is the target text in the decoded image, which is similar to the definition of the MT or seq2seq tasks. Thanks for your suggestion, we have added the explanation of the variable in the new version to eliminate misunderstandings.
>
> **Question 2:**
>
> > The data construction in this paper just renders texts in the parallel corpora into images with white background, which is quite easy data distribution of text images. The authors mention their method has strong scalability, I wonder whether authors conduct any experiments on other data distribution of text images like changing the font, font colors, background colors, and background images?
>
> We also construct datasets with different font types and font sizes, and our end-to-end models reach similar results, and we will add the results in our new version.
>
> **Question 3:**
>
> > What's the method of converting an RGB image into a grayscale map in your implementation (Line 186-188 in the paper)? There are multiple methods of converting an RGB image into a grayscale map, what's your implementation? PIL.image.convert(“L”) or pixel averaging through channels or any other methods? If the channel conversion is conducted using open-source packages or through existing methods, please provide the conversion formula, reference citation, or footnote with a reference link to the open-source package.
>
> We use transforms.Grayscale() of torchvision package to convert the RGB image into the grayscale map. Thanks for your suggestion, it is better to introduce more detail, and we have added a footnote at Line 188 where we firstly mention grayscale map.
>
> **Question 4:**
>
> > Since the dataset in your work is images with black font and white background of the one-line text. Is the RGB image necessary and do you conduct any ablation study?
>
> We construct images with black font and white background. However, the images are not binarized (only black and white pixels). The shape of images in our constructed dataset is HxWxC (C=3), and we call it RGB image in our paper, and then the RGB image is converted into the grayscale map with shape HxWxC (C=1).
>
> **Question 5:**
>
> > Figure 5 shows the Visualization of segmentation with different iteration times. Each segmentation results have several missing pixels at the bottom-right corner, what’s the reason for these missing pixels? Furthermore, What about the robustness of the proposed segmentation method? When there are Gaussian noise, Salt-and-pepper noise, and other pixel noises, what are the influences of the segmentation results? Have you conducted any analysis experiments? Experiments and analysis should be conducted in the paper to show the generalization of the proposed pixel sequence segmentation.
>
> The pixels at the bottom-right corner are not missing, and they are in the next segmentation of the pixel sequence. The smallest unit of a segment is a pixel, and the length of a segment may not be an integer multiple of the height.
>
> Take 5000 iteration in the Figure 5 as an example, the first segmentation containing the image of 'M', and the pixels at the botton-right corner of 'M' are in the second segmentation (containing the image of 'or').The segmentation is learned using the method introduced in sub-section 4.4, and the visualization is the origin segmentation result without any edition.
>
> We conduct experiments on images with pixel noises, and the results show that pixel noises will influence the performance of segmentation. However, after applying denoising, the performance recovers.
>
> Thanks for your comment, we have added detailed explaination of Figure 5.
>
> **Question 6:**
>
> > In sub-section 6.4, there is no comparison and analysis of end-to-end models with cascade models. As so, error propagation indeed exists in cascade methods, and can your proposed end-to-end methods address this problem? What's the relationship between the analysis of error propagation of the cascade method with your proposed end-to-end method? Furthermore, the case study in Table 5 lacks the results of your proposed end-to-end model, which cannot prove your end-to-end model can reduce the influence of error propagation.
>
> The BLEU score and COMET score of our end-to-end model are higher than the cascade method on test sets (Table 1), which demonstrates the end-to-end model has a better translation quality. The error propagation of cascade method is between OCR and NMT models (sub-section 6.4). Specifically, the error in the OCR results will have an influence on the BPE results, and will damage the NMT performance furthermore.  However, for an end-to-end model, there is no passing of results and cascading of models, which does not have the phenomenon of error propagation.
>
> We also check the output image of our end-to-end model with the input image of Table 5, which contains *"The perpetrators also shot a bus stop."*. Compared with the output of the cascade method (*"The perpetrators shot* *Berdem* *on a bus stop ."*), our end-to-end model mitigates the error propagation.
>
> Thanks for your comment, we have added the output of our end-to-end model to Table 5.
>
> **Question 7:**
>
> > For image encoding and decoding, the pixel sequence transformation and pixel sequence segmentation might be one of your contributions. However, there is no comparison with vanilla patch-based image encoding and decoding methods, which is a quite vital baseline for your work.
>
> The patch-based model (Vision Transformer, ViT) is a widely used vision model, which is a vital baseline. The previous work [1] used vision model (CNN), and the performance of our model is better than their model (Table 2).
>
> Furthermore, the shape of our input images is not fixed (HxWxC, W is proportional to the length of the text), and patch-based model and other conventional vision models usually take image with fixed shape as input (for example, the shape of input images of [1] is 32x1024). Thanks for your reminder, we will add this baseline in the new version.
>
> **Missing References:**
>
> > Path-based image encoding should be considered as a vital baseline for this paper: [1] Alexey Dosovitskiy, Lucas Beyer, Alexander Kolesnikov, Dirk Weissenborn, Xiaohua Zhai, Thomas Unterthiner, Mostafa Dehghani, Matthias Minderer, Georg Heigold, Sylvain Gelly, Jakob Uszkoreit, Neil Houlsby: An Image is Worth 16x16 Words: Transformers for Image Recognition at Scale. ICLR 2021
>
> Thanks for your reminder, we have added it in our new version.
>
> **Presentation Improvements:**
>
> > What's the meaning of T in Equation (3)? Meanwhile, the detailed clarification of function F_{ITS}(x) is missing. Subsection 4.2 only introduces 1) transforming the RGB image into a grayscale map and 2) quantizing values in the grayscale map into a character map. I believe that there must have a step 3) transforming character map into pixel sequences but it is missing in the paper. If there is no step 3), the character map of length H\times W (Line 203 in the paper) is different from x_{tok}\in\mathbb{R}^{T\times 1} as in Equation (3).
>
> T is the length of pixel sequence, which is equal to HxW. x is the input RGB image, and x_{tok} is the pixel sequence corresponding to x, F_{ITS}(x) is the mapping form x to x_{tok}.  In sub-section 4.2, we describe the process of mapping.
>
> Like you said, there is a step transforming character map into pixel sequence (step 3). At Line 202-Line 203, we describe the character map that is converted into a sequence of length HxW.  We may not describe this step clearly. Specifically, the character map is organized as a sequence from top to bottom and then from left to right, and the character map with a size of HxW is converted into the 1-D sequence of length T (T=HxW). Thanks for your question, and we have described this step more clearly in the new version.
>
> Thanks for your comments to help us improve our work, and we will continue to polish the paper to make it better.
>
> [1] Mansimov E, Stern M, Chen M, et al. Towards end-to-end in-image neural machine translation[J]. arXiv preprint arXiv:2010.10648, 2020.

---

### Meta-Review · Area_Chair_xuZL · 2023-09-19

**Recommendation:** 4

**Metareview:**

The authors present a pixel sequence-to-sequence model that allows them to translate text in images with an end-to-end model, instead of a cascade between OCR and translation. Experiments show improvements over the cascade on both in- and out-of-domain test sets. The reviewers initially had many concerns with the presentation and experiments, but it seems like most of these have been resolved during the response. The most pressing remaining concern is the realism of the training and test sets, which are formed from black text on a white background, calling into question the generality of the in-image translations that are tested in this paper. The authors correctly point out that, as this is a new task, there are no better datasets, and that this does address a common scenario of images consisting primarily of text. But all three reviewers still seem somewhat disappointed. Given the framing of the paper, it would be much stronger with more realistic data.

---

### Decision · Program_Chairs · 2023-10-07

**Decision:**

Accept-Findings

**Comment:**

The authors present a pixel sequence-to-sequence model that allows them to translate text in images with an end-to-end model, instead of a cascade between OCR and translation. Experiments show improvements over the cascade on both in- and out-of-domain test sets. The reviewers initially had many concerns with the presentation and experiments, but it seems like most of these have been resolved during the response. The most pressing remaining concern is the realism of the training and test sets, which are formed from black text on a white background, calling into question the generality of the in-image translations that are tested in this paper. The authors correctly point out that, as this is a new task, there are no better datasets, and that this does address a common scenario of images consisting primarily of text. But all three reviewers still seem somewhat disappointed. Given the framing of the paper, it would be much stronger with more realistic data.